# Relationships between climate and phylogenetic community structure of fossil pollen assemblages are not constant during the last deglaciation

**Kavya Pradhan[¤a]\*, Diego Nieto-Lugilde[¤b], Matthew C. Fitzpatrick**

University of Maryland Center for Environmental Science, Appalachian Lab, Frostburg, Maryland, United States of America

¤a Current address: Department of Biology, University of Washington, Seattle, Washington, United States of America
¤b Current address: Departamento de Botánica, Ecología y Fisiología Vegetal, Universidad de Córdoba, Córdoba, España
\* kavyap2@uw.edu

**Data Availability Statement:** The fossil pollen data are available from the Dryad Digital Repository (https://doi.org/10.5061/dryad.hk400). The climate data are available from the Dryad Digital Repository

## Abstract

Disentangling the influence of environmental drivers on community assembly is important to understand how multiple processes influence biodiversity patterns and can inform understanding of ecological responses to climate change. Phylogenetic Community Structure (PCS) is increasingly used in community assembly studies to incorporate evolutionary perspectives and as a proxy for trait (dis)similarity within communities. Studies often assume a stationary relationship between PCS and climate, though few studies have tested this assumption over long time periods with concurrent community data. We estimated two PCS metrics—Nearest Taxon Index (NTI) and Net Relatedness index (NRI)—of fossil pollen assemblages of Angiosperms in eastern North America over the last 21 ka BP at 1 ka intervals. We analyzed spatiotemporal relationships between PCS and seven climate variables, evaluated the potential impact of deglaciation on PCS, and tested for the stability of climate-PCS relationships through time. The broad scale geographic patterns of PCS remained largely stable across time, with overdispersion tending to be most prominent in the central and southern portion of the study area and clustering dominating at the longitudinal extremes. Most importantly, we found that significant relationships between climate variables and PCS (slope) were not constant as climate changed during the last deglaciation and new ice-free regions were colonized. We also found weak, but significant relationships between both PCS metrics (i.e., NTI and NRI) and climate and time-since-deglaciation that also varied through time. Overall, our results suggest that (1) PCS of fossil Angiosperm assemblages during the last 21ka BP have had largely constant spatial patterns, but (2) temporal variability in the relationships between PCS and climate brings into question their usefulness in predictive modeling of community assembly.

(http://dx.doi.org/10.5061/dryad.1597g). Data derived from the fossil pollen and climate datasets and associated scripts used in our analyses are available on GitHub (https://github.com/fitzLab-AL/paleoPCS_plosOne). Interested researchers can replicate our study findings in their entirety by either obtaining the original fossil pollen and climate datasets and following the protocol in the Methods section or by using the derived data and scripts available from GitHub.

**Funding:** This research was supported by the National Science Foundation (www.nsf.gov) award number DEB-1257164 to MCF. The funders had no role in study design, data collection and analysis, decision to publish, or preparation of the manuscript.

**Competing interests:** The authors have declared that no competing interests exist.

## Introduction

Determining how abiotic and biotic factors influence biological communities is important for understanding community assembly processes and predicting responses to global change. Studies of community dynamics underscore the prominent role of abiotic factors such as climate in mediating assembly processes and in determining community composition via physiological controls on species occurrence (known as environmental filtering; [1–4]). The paleoecological record provides a window into community dynamics across extended periods of climatic change during which groups of co-occurring taxa disassembled and reassembled, sometimes forming no-analog communities [5]. For vegetation, fossil pollen records from sediment cores suggest an influence of climate on plant taxon associations [6] and show clear linkages between climate and community dynamics, such as synchrony between climatic events and compositional changes in plant assemblages [7, 8].

Although climate plays a primary role in shaping communities, biotic processes such as interactions between species and dispersal also are important. For example, dispersal lags associated with post-glacial migration have been shown to influence the geographic ranges of some European plant species and therefore plant community composition [9, 10]. Biotic interactions such as competition and facilitation can work in concert with environmental filtering and dispersal constraints to influence community structure [11]. Although the effects of biotic interactions are considered to be most prominent at local scales, their influence has been inferred at the scales of species geographic ranges and other macroecological patterns [12–15].

Static observations provide limited information for differentiating the processes that generate patterns of community composition, but Phylogenetic Community Structure (PCS) often has been used for this purpose. PCS quantifies the extent to which co-occurring species are phylogenetically related and can serve as a proxy for trait (dis)similarity to infer the relative importance of abiotic and biotic processes in determining community structure [16]. PCS patterns can be compared to random assemblages to test whether the degree of relatedness of co-occurring species differs from that expected by chance [17, 18], with nonrandom patterns typically being interpreted as reflecting the outcome of either environmental filtering or competition. Environmental filtering generally is hypothesized to lead to clustered communities (co-occurring species are more related / have greater trait similarity than expected by chance; [19, 20]), while competition should tend to lead to overdispersed communities (co-occurring species are less related / have more dissimilar traits than expected by chance; [18, 19]). It is important to note however that these general predictions for relationships between nonrandom patterns of PCS and the processes that generate patterns of community composition do not always hold given (1) the complex interplay between climate, phylogeny, and biotic processes (mediated through functional traits) and (2) that a particular set of processes can generate multiple patterns, and vice versa [21–24]. In addition, PCS can be influenced by processes other than competition and contemporary environmental filtering, notably the legacy of past climatic conditions due to dispersal lags [25] or intercontinental migrations and in-situ speciation [26].

In addition to complexities that can obscure links between processes and patterns using PCS, studies typically assume that PCS-climate relationships are stable through time, and although community assembly is a dynamic process [8], most studies have analyzed PCS using static community data [27–29]. When studies have examined PCS through time, most have either used (1) a proxy for time rather than a time series of observations (e.g., chronosequences; [30–33] or (2) relatively short time sequences [34–36]. Studies that have analyzed temporal patterns in PCS most often have found an increase in phylogenetic overdispersion (increasing role of competition) through time [30, 31, 33, 36], while others have found

evidence for increased phylogenetic clustering (increasing role of environmental filtering) through time [35].

Extensive paleoecological time series provide an opportunity to examine assemblage dynamics across large spatial extents and extended periods of time as species responded to climatic events, which, when combined with analyses of PCS, may provide inferences regarding the processes shaping community composition that otherwise may be difficult to obtain from analyses of static patterns. In this study, we examine spatiotemporal patterns of fossil pollen assemblages of Angiosperms across eastern North America from the Last Glacial Maximum (LGM; 21 ka BP—thousands of years before present) to present (0 BP) to assess how PCS evolves as previously glaciated areas are colonized. Specifically, we address three questions: (1) Does PCS of fossil Angiosperm assemblages exhibit nonrandom patterns across space and through time since the LGM?; (2) What is the relationship between PCS and climate, and does this relationship remain stable through time?; and (3) How does deglaciation impact PCS through its influence on subsequent colonization and succession processes?

We expected to find a significant effect of climate on PCS of pollen assemblages, with phylogenetically clustered communities in places and at times where environmental filtering should dominate, namely cold high latitudes, semi-arid regions, and harsh LGM climates. In contrast, we expect phylogenetically overdispersed communities in places and at times where the role of environmental filtering should be reduced, such as warm low latitudes, moist regions, and benign present-day climates. Furthermore, if the PCS-climate relationship is one of the main assembly rules for communities, we predicted that it should remain stable through time. Finally, given the importance of glaciations and the potential impacts of postglacial migrations on plant species and communities, we expected to find a strong signal of time-since-deglaciation on PCS. Our analyses revealed a persistent geographic pattern of phylogenetic overdispersion in the central and southern portion of the study region. Most importantly, relationships between PCS and both climate and time-since-deglaciation varied through time.

## Material and methods

### Pollen occurrence data

We focus on the last deglaciation (21 ka BP to present) of the eastern and southern half of North America (113°30´ - 53°00´W / 25°00´ - 61°00´N), where time series of fossil pollen records are relatively dense. We used taxon occurrence records from 21 ka BP to 0 ka BP (present) at 1 ka intervals as described by [37, 38]. Paleoecological records suffer from a host of uncertainties and vary greatly in their degree of temporal and taxonomic resolution. We minimized these issues to the greatest extent possible by selecting fossil-pollen records that (1) have low temporal uncertainty (< 500 years at most sites), (2) use a standardized taxonomy for identification, and (3) estimate occurrence using robust temporal interpolations of relative abundance expressed as the pollen sum for a particular taxon divided by the total sum for all genus-level taxa, rather than divided by the total upland sum for the site (which includes both genera as well as families and other higher level taxa; [39]. Pollen data quality was calculated as in [37, 38], and for each 1 ka time step, only sites with a weighted quality value above 0.75 were included. If multiple sites fell within the same 0.5 x 0.5 degrees grid cell (see Climate and Ice Sheet Data section), their pollen abundances were averaged. The majority of grid cells contained pollen assemblages from within a single lake sediment core. We used pollen relative abundances for each climate grid cell and 1 ka time interval. The phylogenetically related taxa pairs *Oxyria-Rumex*, *Juniperus-Thuja*, and *Ostrya-Carpinus* cannot readily be distinguished from pollen. Hence, those taxa were combined to form a unique branch in the phylogenetic tree (see Phylogenetic Data section). Additionally, *Ambrosia*-type includes multiple genera in

Asteraceae that have similar pollen morphology that are generally indistinguishable. Nonetheless, within the Asteraceae *Iva* and *Xanthium* are distinguished in the pollen dataset and were treated independently. However, close relatives have similar pollen morphology and therefore such aggregation should not bias our results.

All told, the initial dataset included 106 pollen taxa identified to the genus level of both Gymnosperms and Angiosperms. We decided to run analyses only with Angiosperms (n = 96) to prevent deep splits in the phylogenies from confounding PCS metrics (see below) and because the low number of Gymnosperm taxa (n = 10) prevented estimation of PCS for most of the locations and time slices when Gymnosperms are examined separately. This resulted in 96 Angiosperms taxa at the genus level (S1 Fig). Pollen data are available from the Dryad Digital Repository https://doi.org/10.5061/dryad.hk400 [38].

## Climate and ice sheet data

Climate data for North America were obtained from CCSM3 transient simulation [40] and were subsequently processed and downscaled to yearly, quarterly, and monthly variables at a 0.5 x 0.5 degree grid and 1 ka intervals by [41]. We selected seven uncorrelated variables (Pearson < 0.75) that capture the interplay of water availability and energy, and therefore should influence the taxonomic composition of vegetation, including: minimum temperature of the coldest month (Tmin), maximum temperature of the warmest month (Tmax), minimum precipitation of the driest month (Pmin), maximum precipitation of the wettest month (Pmax), mean yearly actual evapotranspiration (AET), mean yearly ratio of actual and potential evapotranspiration (ETR), and mean yearly water deficit index (WDI). Climate data are available from the Dryad Digital Repository http://dx.doi.org/10.5061/dryad.1597g [41].

To study the effect of postglacial migration on PCS, maps of ice sheet extent at each 1 ka time interval were obtained from [42]. Because of minor temporal mismatches between the ice sheet maps and our climate and pollen datasets, we assigned each ice sheet layer to the closest 1 ka time interval in our data. Using these ice sheet maps, we classified cells during each of the 1 ka time intervals as either glaciated or deglaciated and then calculated the time since each grid cell was deglaciated (DEGLAC) at each time period. We identified 9 instances in which a grid cell at the same time period both contained a pollen site and also was considered to be glaciated. These cells were always adjacent to the border of the ice sheet layer and therefore most likely arose due to temporal mismatches between the ice sheet layers and the pollen data and/ or low spatial precision of the ice sheet layers. Given the higher integrity of the pollen data, for these few instances we reclassified such conflicting cells as deglaciated.

## Phylogenetic data

To obtain a phylogenetic tree for the pollen taxa, we used the phylogeny from the megatree Open Tree of Life (OToL v.12.3; [43]). This megatree provides a topology for the tree of life based on multiple phylogenetic studies, resolving unstudied branches using known and accepted taxonomy (i.e., based on non-phylogenetic data). [44] have recently shown that OToL megatree provides the most similar results to purpose-built phylogenies compared to other available megatrees. We queried OToL for Spermatophyta (seed plants) using the taxonomic names of the pollen taxa at the genus level, retaining the 109 taxa in the initial pollen dataset (106 taxa after combining indistinguishable pollen taxa). Mismatches between pollen taxonomy and OToL taxonomy were resolved manually (e.g. *Prunus* pollen type is Amygdaleae in OToL; S1 Fig). To add branch lengths, which OToL does not provide, we obtained ages from the datelife project (http://datelife.org/; [45] for as many nodes as possible (n = 36 of 98 nodes; S1 Fig), including Gymnosperm taxa. Lastly, we used the BLADJ algorithm in

Phylocom [46] to estimate ages for the remaining nodes, thereby providing an estimate of branch lengths for the entire tree. We removed Gymnosperm taxa after calculating branch lengths, because the old split between Gymnosperms and Angiosperms helps to constrain node ages for the entire tree. Phylogenetic data are available from the Open Tree of Life https://blog.opentreeoflife.org [43]. We used the 'rotl' [47] and 'datelife' packages [48] in R [49].

## Analysis

We calculated both Nearest Taxon Index (NTI) and Net Relatedness Index (NRI) to quantify PCS in each occupied grid cell and at each 1 ka time slice using the 'Picante' package [50] in R [49]. NRI and NTI represent different components of evolutionary history [17, 18]. NRI is calculated based on mean phylogenetic distance, which takes into account deep phylogenetic divergences, while NTI is calculated on the basis of mean nearest taxon distance, which takes more recent divergences into account. To standardize effect size for both metrics (i.e., NRI and NTI), the observed values were compared to a distribution of values from the "independentswap" null model (with default parameters), which randomizes the community data matrix while maintaining species occurrence frequency and sample species richness [51]. Note that we also performed preliminary sensitivity analyses using all null models available in 'Picante'. Although each null model provided a completely different set of values of NTI and NRI the main conclusions of our study remained unchanged.

We mapped patterns of NTI and NRI at each fossil pollen site in each 1 ka time step and used Inverse Distance Weighting to interpolate to unmeasured locations. We used ordinary least square (OLS) regression to quantify the relationship between NTI and NRI and each of the climatic drivers and time-since-deglaciation (e.g. NRI ~ TMIN). PCS metrics exhibited spatial autocorrelation and for many variables were not homoscedastic. Therefore, we fitted preliminary models using Quantile Regression (QR) and error based Spatial Autoregressive (SAR) models. Because OLS and QR models exhibited spatial autocorrelation in the residuals and QR models did not provide additional insights, we proceeded with fitting only OLS models, which are easier to interpret, and SAR models, which fix bias in parameter estimation. The SAR models were fit using the 'spdep' package [52] in R [49].

Because we were interested in testing whether or not PCS-climate relationships were stable through time, we fit several OLS and SAR models with three levels of complexity for each climate variable: (1) Stable-Relationship (PCS ~ climate), which assumes that the relationship between PCS and climate remained constant through time; (2) Stable-Slope (PCS ~ climate + time); which assumes that the slope of the relationship was stable through time but the intercept may have changed; and (3) Changed-Relationship (PCS ~ climate*time), which assumes that both the slope and the intercept may have changed through time. In the Stable-Relationship models, time is neglected, whereas in the Stable-Slope models time is included as an additional variable. In the Changed-Relationship models time is included as a variable interacting with climate. We opted to assess each variable separately because conducting a multiple regression would complicate our ability to disentangle the effect of each variable. We compared the three versions of the OLS and SAR models using ANOVA. The most supported model in the ANOVA provides inference on whether the intercept and/or the slopes of PCS-climate relationships have changed through time. To avoid biases caused by low sample sizes, time periods that had less than 10 sites with PCS values were excluded (i.e., 17–21 ka BP). SAR models were fit using three different distance-based weight matrices: 120, 360, and 480 km. For each climate variable, we selected the distance that resulted in the lowest residual Moran's I values when fitting the Stable-Relationship model. The selected distances were then used to fit the

Stable-Slope and Changed- Relationship SAR models. To study the effect of postglacial migration on PCS, we additionally plotted the average PCS values across all cells of the same time-since- deglaciation for each time period.

## Results

### Spatiotemporal patterns in PCS

Across all twenty-two 1 ka time steps, plant assemblages recorded in fossil pollen exhibited NRI values (Fig 1a) ranging from -3.00 (overdispersion) to 2.86 (clustering), and NTI values (Fig 1b) ranging from -3.42 to 2.34. Both NRI and NTI exhibited variation across space, with positive values (clustering) tending to occur in the east (slope = 0.015, $r^2$ = 0.078, p < 0.001 for NRI; slope = 0.015, $r^2$ = 0.054, p < 0.001 for NTI) and north (slope = 0.019, $r^2$ = 0.037, p < 0.001 for NRI; slope = 0.020, $r^2$ = 0.025, p < 0.001 for NTI) of the study area based on OLS regressions of NRI and NTI against latitude and longitude. Although the range of NRI and NTI values varied through time, the broad scale geographic pattern remained largely constant (Fig 1).

### Climate-PCS relationship

Most OLS models relating NRI or NTI with climate variables were significant (Table 1). Among the OLS models with stable slope and intercept through time (i.e., Stable- Relationship), NRI had a positive relationship with Pmin, Pmax, AET, ETR, and WDI, and a negative relationship with Tmin and Tmax (Table 1, Fig 2 and S2 Fig). NTI had a negative relationship with all variables except for Pmin, ETR and WDI (Table 1, S3 Fig). For the Stable-Slope models, all relationships were similar to Stable-Relationship models, except for NRI~AET which had a negative relationship. For the Changed-Relationships model, the intercepts and slopes for both NRI and NTI varied through time (Figs 3 and 4) and there was no consistent trend in the fluctuation between positive and negative slopes for either PCS metric. However, all OLS models, including those for Changed-Relationship, explained only a small amount of variation in PCS (mean adjusted $r^2$ = 0.028 and 0.019 SD for NRI and mean adjusted $r^2$ = 0.043 and 0.024 SD for NTI; Table 1). Variance explained tended to increase from the simplest model (Stable-Relationship models) to the most complex (Changed-Relationship models; Table 1).

The raw NRI and NTI values exhibited significant spatial structure (S1 Table) at all three neighborhood distance classes (i.e., 120, 360, 480 km), and correspondingly, all of the OLS models showed significant spatial autocorrelation in model residuals (S2 Table). SAR models effectively removed spatial autocorrelation at all three neighborhood distance classes (S3 Table), and for both NRI and NTI, SAR models fitted using a neighborhood of 480 km and 360 km, respectively, were most parsimonious (lowest AIC; S4 Table). Here forward we report results only for these SAR models and indicate explicitly when referring to OLS models.

When spatial autocorrelation was removed, the significance levels of the regression models for both NRI and NTI generally decreased and, in some cases, became non-significant (Tmin, AET and ETR for NRI; and Tmax, Pmax, and ETR for NTI; Table 1). Among the SAR models with a stable slope through time (i.e., Stable-Relationship and Stable-Slope), NRI showed a positive correlation with all climate variables other than Tmin, whereas NTI showed a negative correlation with all climate variables and models types except ETR and Pmin in the Stable-Relationship model, and ETR, Pmin, and Tmax in the Stable-Slope model (Table 1). Similar to OLS models, in the Changed-Relationships SAR models, the intercepts and slopes for both NRI and NTI varied through time (Figs 3 and 4) with no clear trend towards more positive or more negative slopes.

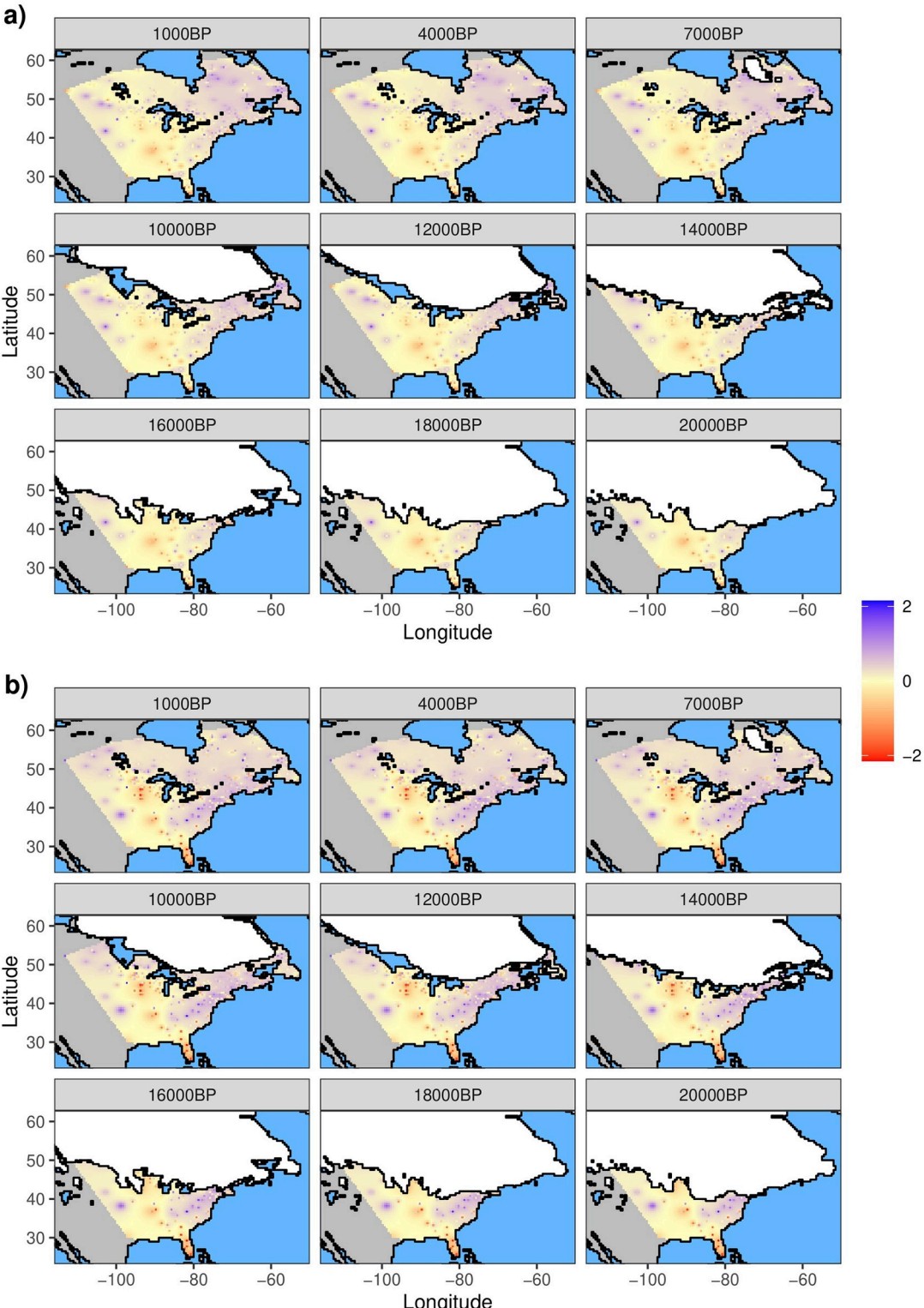

**Fig 1. Spatial pattern of phylogenetic community structure through time.** a) Net Relatedness Index (NRI) and b) Nearest Taxon Index (NTI) for the study area through time (for clarity only a subset of time periods are plotted). Maps were calculated using Inverse Distance Weighted interpolation using actual NRI and NTI values from cells with pollen occurrences. Higher values (purple shading) indicate more clustered communities (organisms are more related) and lower values (red shading) indicate overdispersed communities (organisms are less related). Ice sheet extent at each time period is represented with a white polygon.

**Table 1. Parameters of ordinary least square regression (OLS) and spatial autoregressive (SAR) models relating two metrics of phylogenetic community structure (PCS; net relatedness index—NRI—And nearest taxon index—NTI) with seven climate variables.** Each combination of PCS metric and climate variable was modeled three times, allowing distinct levels of variation to study the evolution of the parameters through time: stable-relationship, stable-slope, and changed relationship. SAR models reported here were fit selecting neighbors at distances of 480 km for NRI and 360 km for NTI.

| PCS metric | Model type | Var. | Stable-Relationship | | | | Stable-Slope | | | Changed-Relationship | |
|---|---|---|---|---|---|---|---|---|---|---|---|
| | | | Int. | Slope | Adj. R² | P val. | Slope | Adj. R² | P val. | Adj. R² | P val. |
| NRI | OLS | Tmin | 0.06 | -0.007 | 0.012 | *** | -0.009 | 0.03 | *** | 0.09 | *** |
| | | Tmax | 0.769 | -0.024 | 0.041 | *** | -0.026 | 0.059 | *** | 0.103 | *** |
| | | Pmin | -0.004 | 0.004 | 0.026 | *** | 0.004 | 0.038 | *** | 0.063 | *** |
| | | Pmax | 0.232 | 0 | 0 | ns | 0 | 0.012 | * | 0.051 | *** |
| | | AET | 0.469 | 0 | 0.032 | *** | -0.001 | 0.055 | *** | 0.107 | *** |
| | | ETR | -0.397 | 0.657 | 0.028 | *** | 0.673 | 0.04 | *** | 0.052 | *** |
| | | WDI | 0.077 | 0 | 0.057 | *** | 0.001 | 0.088 | *** | 0.103 | *** |
| | | DEGLAC | 0.362 | 0 | 0.021 | *** | 0 | 0.061 | *** | 0.085 | *** |
| | SARerr (480 km) | Tmin | 0.155 | 0 | 0.143 | ns | -0.004 | 0.149 | ns | 0.214 | *** |
| | | Tmax | -0.252 | 0.012 | 0.146 | *** | 0.023 | 0.152 | *** | 0.186 | *** |
| | | Pmin | -0.064 | 0.005 | 0.149 | *** | 0.004 | 0.153 | *** | 0.182 | ns |
| | | Pmax | 0.042 | 0.001 | 0.144 | ns | 0 | 0.148 | ns | 0.182 | ** |
| | | AET | 0.182 | 0 | 0.143 | ns | 0 | 0.15 | * | 0.206 | *** |
| | | ETR | -0.007 | 0.202 | 0.144 | ns | 0.157 | 0.148 | ns | 0.161 | ns |
| | | WDI | 0.159 | 0 | 0.147 | *** | 0 | 0.153 | *** | 0.165 | * |
| | | DEGLAC | 0.204 | 0 | 0.144 | ns | 0 | 0.154 | *** | 0.19 | *** |
| NTI | OLS | Tmin | 0.006 | -0.012 | 0.022 | *** | -0.012 | 0.028 | *** | 0.047 | *** |
| | | Tmax | 0.885 | -0.027 | 0.034 | *** | -0.03 | 0.049 | *** | 0.061 | *** |
| | | Pmin | -0.14 | 0.007 | 0.06 | *** | 0.007 | 0.07 | *** | 0.081 | *** |
| | | Pmax | 0.578 | -0.003 | 0.013 | *** | -0.004 | 0.027 | *** | 0.064 | *** |
| | | AET | 0.651 | -0.001 | 0.048 | *** | -0.001 | 0.056 | *** | 0.078 | *** |
| | | ETR | -0.692 | 1.029 | 0.044 | *** | 1.05 | 0.054 | *** | 0.062 | *** |
| | | WDI | 0.057 | 0.001 | 0.083 | *** | 0.001 | 0.097 | *** | 0.105 | *** |
| | | DEGLAC | 0.478 | 0 | 0.028 | *** | 0 | 0.034 | *** | 0.043 | *** |
| | SARerr (360 km) | Tmin | 0.029 | -0.01 | 0.226 | *** | -0.003 | 0.233 | ns | 0.254 | *** |
| | | Tmax | 0.324 | -0.006 | 0.221 | ns | 0.001 | 0.233 | ns | 0.249 | *** |
| | | Pmin | 0.071 | 0.003 | 0.222 | * | 0.005 | 0.236 | *** | 0.244 | ns |
| | | Pmax | 0.302 | -0.001 | 0.222 | ns | -0.003 | 0.236 | *** | 0.255 | *** |
| | | AET | 0.456 | 0 | 0.229 | *** | 0 | 0.235 | ** | 0.264 | *** |
| | | ETR | -0.031 | 0.263 | 0.222 | ns | 0.191 | 0.233 | ns | 0.241 | ns |
| | | WDI | 0.183 | 0 | 0.225 | *** | 0 | 0.234 | ns | 0.245 | ns |
| | | DEGLAC | 0.361 | 0 | 0.232 | *** | 0 | 0.236 | ** | 0.247 | *** |

*** $p<0.001$;

** $p<0.01$;

* $p<0.05$.

ns non-significant ($p>0.05$)

Tmin = minimum temperature of the coldest month; Tmax = maximum temperature of the warmest month;

Pmin = minimum precipitation of the driest month; Pmax = maximum precipitation of the wettest month;

AET = mean yearly actual evapotranspiration; ETR = mean yearly ratio of actual and potential evapotranspiration;

WDI = mean yearly water deficit index; DEGLAC = time-since-deglaciation.

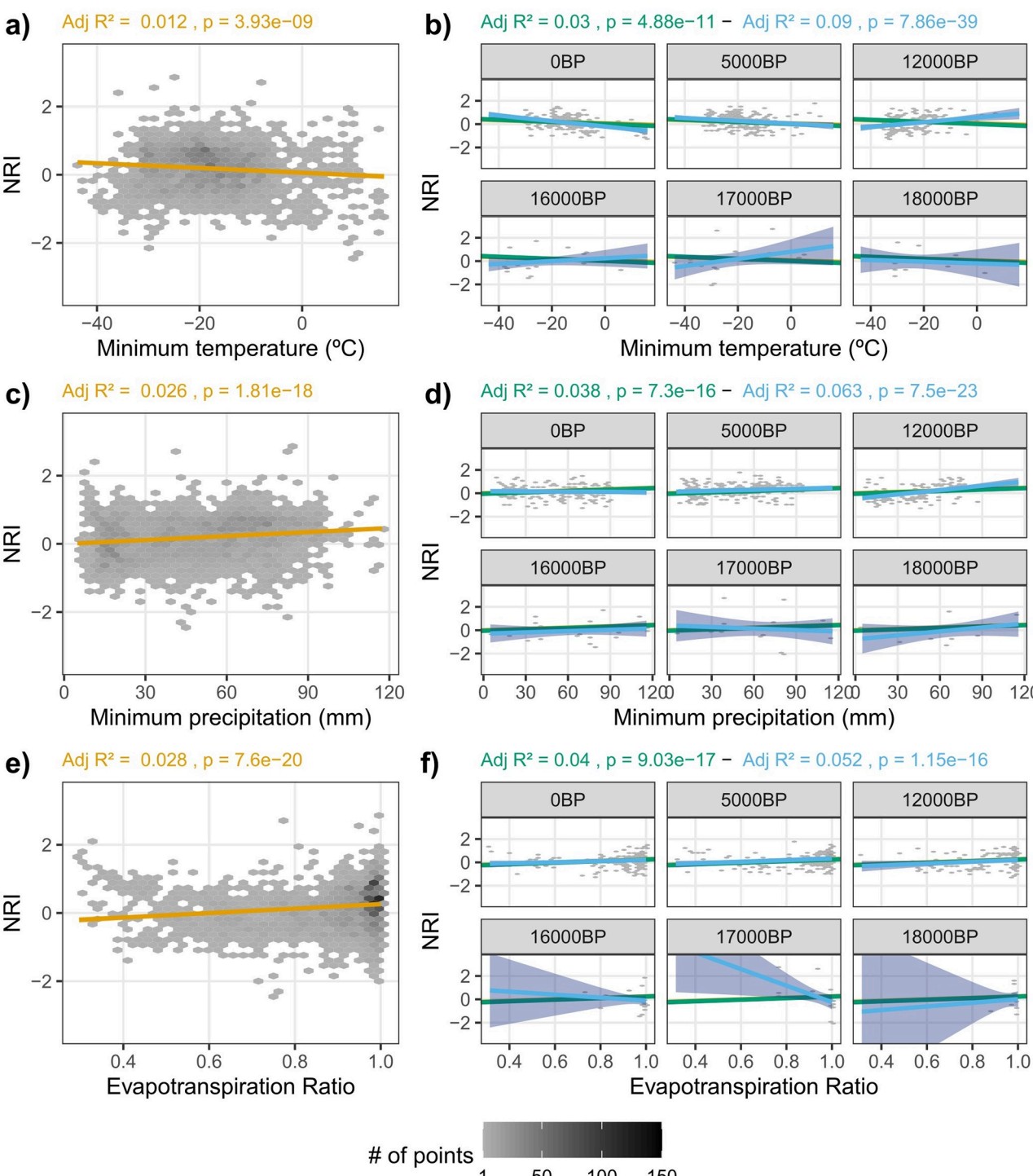

**Fig 2. Relationship between net relatedness index (NRI) and three climate variables and their change through time according to the three fitted models.** Gray shading in the scatter plots represents the count of points falling in each bin (hexagons). The panels on the left (a, c, e) represent the overall relationship according to ordinary least square regression when pooling all time periods (Stable-Relationship; orange lines). The panels on the right (b, d, f) show the relationship between NRI and climate variables as estimated with the data for a subset of time periods, with green lines representing Stable-Slope Model, blue lines representing Changed-Relationship Model, and the orange line showing the overall relationship from panels on the left for comparison. Note that each model is fitted to data for all time periods and hence a single adjusted $R^2$ and p-value for each model is presented in the same colors as the lines. Shaded areas represent the confidence intervals at 95% for the regression lines.

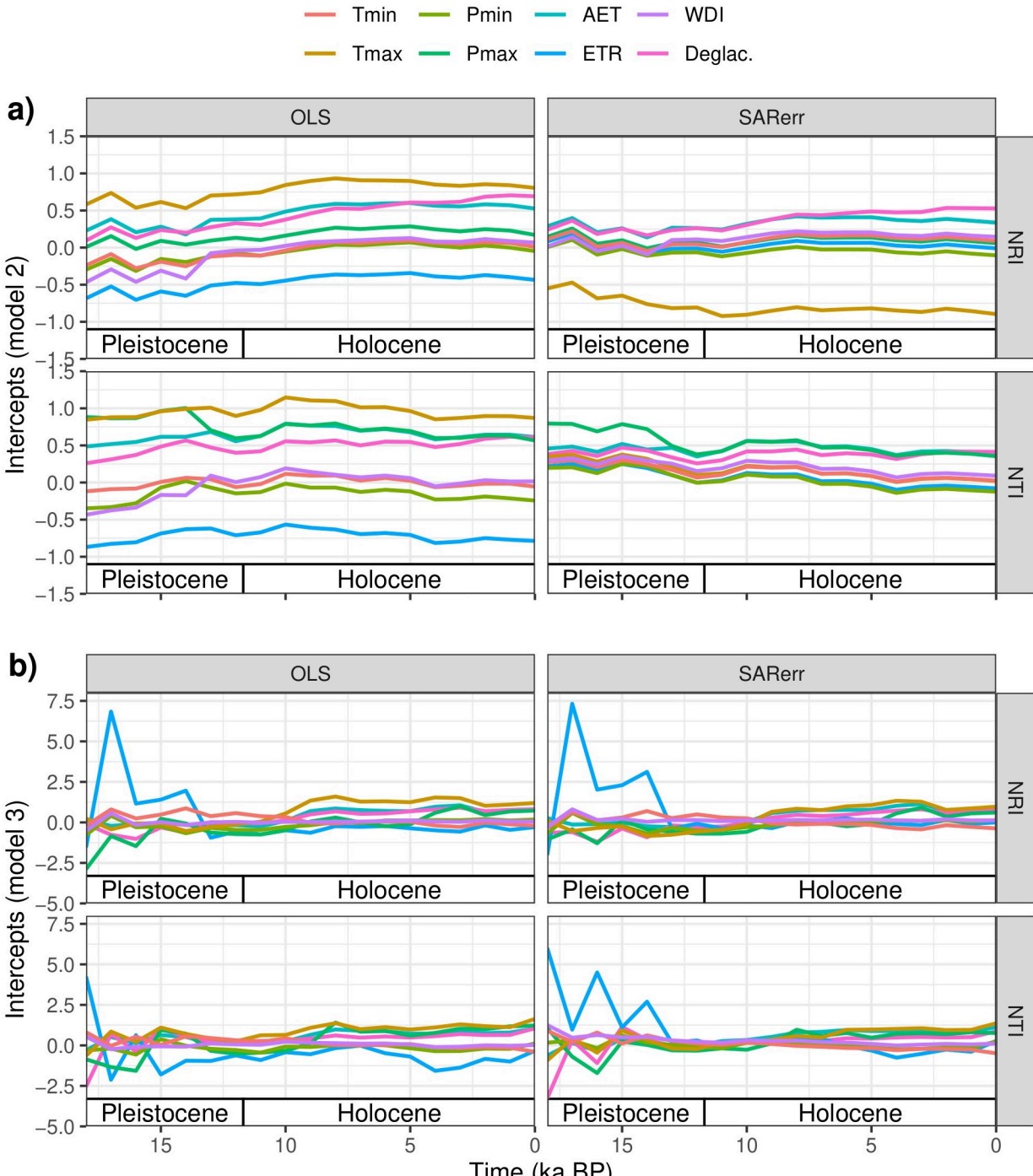

**Fig 3. Model intercepts through time.** Intercepts of OLS and SAR models for each variable plotted across time for a) Stable-Slope and b) Changed-Relationship models. Tmin = minimum temperature of the coldest month; Tmax = maximum temperature of the warmest month; Pmin = minimum precipitation of the driest month; Pmax = maximum precipitation of the wettest month; AET = mean yearly actual evapotranspiration; ETR = mean yearly ratio of actual and potential evapotranspiration; WDI = mean yearly water deficit index; DEGLAC = time-since-deglaciation.

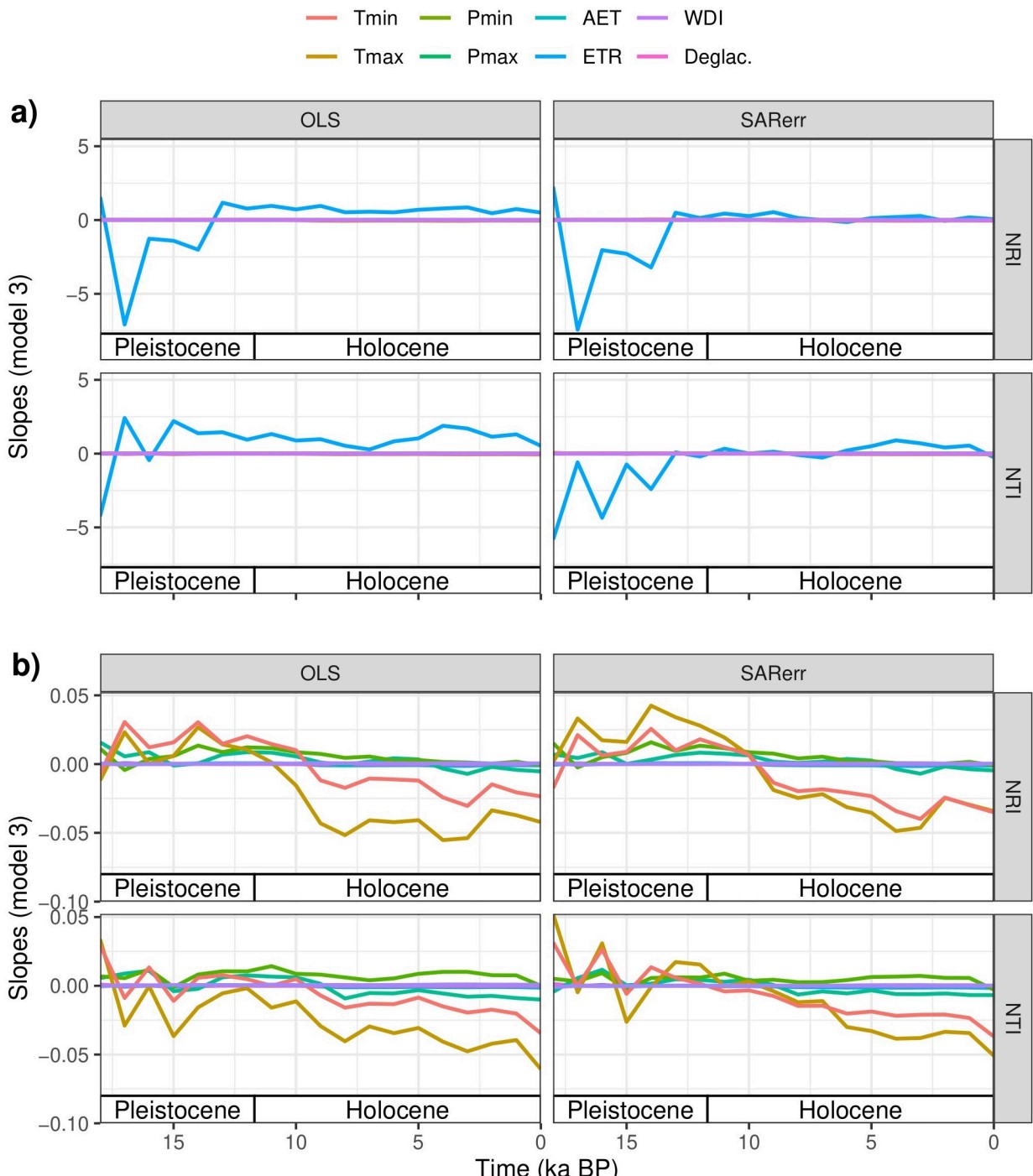

**Fig 4. Model slopes through time.** Slopes of OLS and SAR models for each variable plotted across time for the Changed-Relationship models. Slopes represent increments on the PCS metric by one-unit increments in each independent variable, hence, units differ among variables. Note that for readability a) contains slopes for ETR while b) shows slopes for all other variables as the magnitude of slope values for ETR was much greater than that of other variables. Tmin = minimum temperature of the coldest month; Tmax = maximum temperature of the warmest month; Pmin = minimum precipitation of the driest month; Pmax = maximum precipitation of the wettest month; AET = mean yearly actual evapotranspiration; ETR = mean yearly ratio of actual and potential evapotranspiration; WDI = mean yearly water deficit index; DEGLAC = time-since-deglaciation.

The explanatory power of the models increased appreciably in the SAR models compared to the OLS models and these increases were greater for NTI than NRI models (for NRI: mean adjusted $r^2$ = 0.160 and 0.021 SD across all models; and for NTI: mean adjusted $r^2$ = 0.236 and 0.011 SD; Table 1). SAR models also showed an increase in explained deviance for both PCS metrics among models types, with Stable-Relationship having the lower explained deviance (for NRI: mean adjusted $r^2$ = 0.145, 0.002 SD and for NTI: mean adjusted $r^2$ = 0.225, 0.004 SD), followed by Stable-Slope (for NRI: mean adjusted $r^2$ = 0.151, 0.002 SD and for NTI: mean adjusted $r^2$ = 0.234, 0.001 SD), and Changed-Relationship having the higher value of explained deviance (for NRI: mean adjusted $r^2$ = 0.186, 0.0018 SD and for NTI: mean adjusted $r^2$ = 0.250, 0.007 SD). Lastly, in an *ad hoc* analysis we re-ran all models and ANOVA analysis equalizing the sample size across time periods. The results from this sensitivity analysis showed that models were not biased by differing sample sizes among time periods (S5 and S6 Tables and S4–S7 Figs).

For both NRI and NTI, ANOVA revealed stronger support for Stable-Slope models than Stable-Relationship models for most climate variables (Table 2). However, there were several exceptions to this pattern, including, for NRI, all climate variables with SAR models and, for NTI, for Tmin and AET in both OLS and SAR models, and ETR and WDI in OLS models. Changed-Relationship models also generally had greater support than Stable-Slope models, except for NTI for ETR and WDI in OLS models.

## Impact of deglaciation on PCS

Time-since-deglaciation explained a small amount of variation in OLS models and a greater in SAR models. Furthermore, the amount of variation explained also increased with increasing model complexity (Stable-Relationship to Changed-Relationship). Model selection using ANOVA indicated that the Stable-Relationship model was the least supported for both OLS and SAR analyses for both PCS metrics, with the exception of OLS models for NTI. Intercepts for time-since-deglaciation show similar trends to those observed for climate variables in Stable-Slope and Changed-Relationship models (Fig 3). When aggregating (by averaging) PCS values for each combination of time-since-deglaciation and time period (Fig 5), we found an interaction between time and time-since-deglaciation for NRI, but not for NTI. Sites that had deglaciated more recently (low time-since-deglaciation) were more phylogenetically clustered than sites that had been deglaciated for longer time periods (high time-since-deglaciation), but only for more recent time periods (i.e., from ~ 9 ka BP to present). When we removed the effect of all climate variables using multiple regression models, this pattern, although less distinct, remained (Fig 5).

## Discussion

The goal of our study was to answer three general questions regarding spatiotemporal relationships between climate and PCS using fossil Angiosperm assemblages since the LGM: (1) Are spatiotemporal patterns of PCS nonrandom?; (2) If so, are these relationships stable?; and (3) Is there a signal of declaciation on PCS given the influence of colonization and succession processes on assemblages? Our analyses suggest that changes in vegetation in eastern North America over the last 21 ka have been accompanied by largely consistent, nonrandom spatial patterns of PCS and temporally varied relationships between PCS and climate through time. In terms of spatial patterns of PCS, fossil Angiosperm assemblages tended to be phylogenetically clustered in the northeastern parts of the study area and overdispersed in the central and southern regions during all time periods. In contrast, the relationships between PCS and climate varied through time and exhibited substantial autocorrelation. Finally, we found that the relationship between time-since-deglaciation and PCS also varied through time.

**Table 2. ANOVA-based model selection of ordinary least square regression (OLS) and spatial autoregressive (SAR) models relating net relatedness index (NRI) and nearest taxon index (NTI) with seven climate variables.** Each ANOVA was run for each combination of PCS metric and climate variable comparing three different models that allow distinct levels of variation to study the evolution of the regression parameters through time: stable-relationship, stable-slope, and changed relationship. SAR models reported here were fit selecting neighbors at distances of 360 km.

| PCS metric | Model type | Var. | Stable-Relationship | | Stable-Slope | | Changed-Relationship | | Selected Model |
|---|---|---|---|---|---|---|---|---|---|
| | | | Res df (OLS) / Df (SAR) | F (OLS) / L ratio (SAR) | Res df (OLS) / Df (SAR) | F (OLS) / L ratio (SAR) | Res df (OLS) / Df (SAR) | F (OLS) / L ratio (SAR) | |
| NRI | OLS | Tmin | 2961 | NA | 2943 | 3.23 | 2925 | 10.832 | 2***, 3*** |
| | | Tmax | 2961 | NA | 2943 | 3.343 | 2925 | 8.024 | 2***, 3*** |
| | | Pmin | 2961 | NA | 2943 | 2.216 | 2925 | 4.305 | 2**, 3*** |
| | | Pmax | 2961 | NA | 2943 | 1.954 | 2925 | 6.663 | 2**, 3*** |
| | | AET | 2961 | NA | 2943 | 4.17 | 2925 | 9.361 | 2***, 3*** |
| | | ETR | 2961 | NA | 2943 | 2.108 | 2925 | 2.03 | 2**, 3** |
| | | WDI | 2961 | NA | 2943 | 5.605 | 2925 | 2.8 | 2***, 3*** |
| | | DEGLAC | 2961 | NA | 2943 | 7.189 | 2925 | 4.317 | 2***, 3*** |
| | SARerr (480 km) | Tmin | 4 | NA | 22 | 18.815 | 40 | 236.344 | 3*** |
| | | Tmax | 4 | NA | 22 | 20.278 | 40 | 120.693 | 3*** |
| | | Pmin | 4 | NA | 22 | 12.331 | 40 | 103.117 | 3*** |
| | | Pmax | 4 | NA | 22 | 14.77 | 40 | 119.259 | 3*** |
| | | AET | 4 | NA | 22 | 22.084 | 40 | 202.106 | 3*** |
| | | ETR | 4 | NA | 22 | 15.884 | 40 | 43.22 | 3*** |
| | | WDI | 4 | NA | 22 | 24.061 | 40 | 4.803 | 3** |
| | | DEGLAC | 4 | NA | 22 | 35.165 | 40 | 126.697 | 2**, 3*** |
| NTI | OLS | Tmin | 2961 | NA | 2943 | 1.041 | 2925 | 3.203 | 3*** |
| | | Tmax | 2961 | NA | 2943 | 2.552 | 2925 | 2.195 | 2***, 3** |
| | | Pmin | 2961 | NA | 2943 | 1.75 | 2925 | 1.906 | 2*, 3* |
| | | Pmax | 2961 | NA | 2943 | 2.427 | 2925 | 6.462 | 2***, 3*** |
| | | AET | 2961 | NA | 2943 | 1.469 | 2925 | 3.834 | 3*** |
| | | ETR | 2961 | NA | 2943 | 1.676 | 2925 | 1.485 | 2* |
| | | WDI | 2961 | NA | 2943 | 2.522 | 2925 | 1.444 | 2*** |
| | | DEGLAC | 2961 | NA | 2943 | 1.028 | 2925 | 1.452 | 1 |
| | SARerr (360 km) | Tmin | 4 | NA | 22 | 25.939 | 40 | 81.709 | 3*** |
| | | Tmax | 4 | NA | 22 | 44.585 | 40 | 60.848 | 2***, 3*** |
| | | Pmin | 4 | NA | 22 | 54.835 | 40 | 31.165 | 2***, 3* |
| | | Pmax | 4 | NA | 22 | 54.521 | 40 | 76.093 | 2***, 3*** |
| | | AET | 4 | NA | 22 | 26.589 | 40 | 110.927 | 3*** |
| | | ETR | 4 | NA | 22 | 44.883 | 40 | 31.303 | 2***, 3* |
| | | WDI | 4 | NA | 22 | 33.69 | 40 | 43.924 | 2*, 3*** |
| | | DEGLAC | 4 | NA | 22 | 14.671 | 40 | 46.457 | 3*** |

*** $p < 0.001$;

** $p < 0.01$;

* $p < 0.05$.

Tmin = minimum temperature of the coldest month; Tmax = maximum temperature of the warmest month; Pmin = minimum precipitation of the driest month; Pmax = maximum precipitation of the wettest month; AET = mean yearly actual evapotranspiration; ETR = mean yearly ratio of actual and potential evapotranspiration; WDI = mean yearly water deficit index; DEGLAC = time-since-deglaciation.

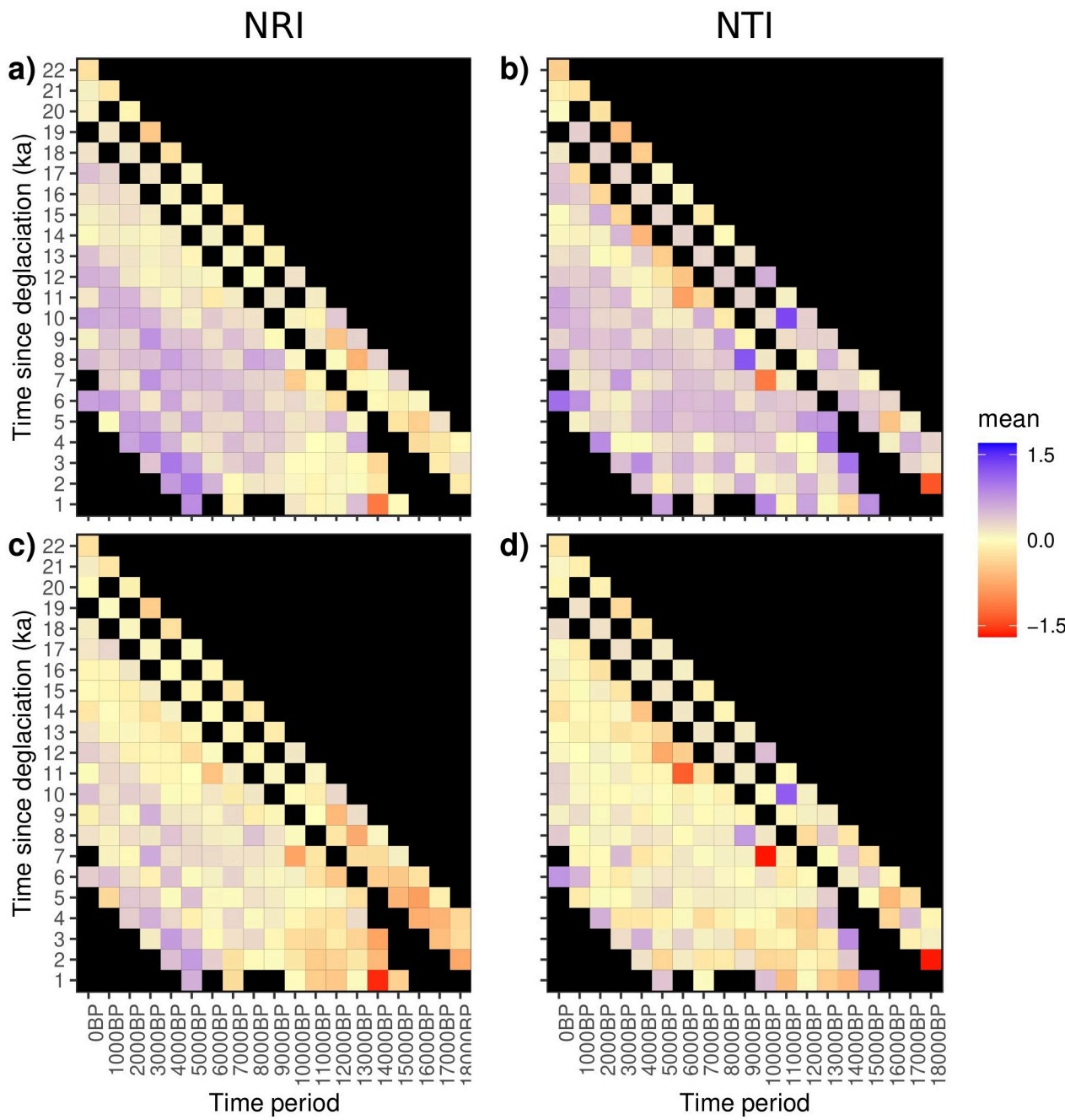

**Fig 5. Pattern of time-since-deglaciation on net relatedness index (NRI) and nearest taxon index (NTI).** Each cell in the heatmap represents the average a) NRI or b) NTI value for cells of a particular time-since-deglaciation at a particular time period. Black squares indicate absence of the particular time-since-deglaciation class for that time period. The second row of heatmaps represent the average of residuals for c) NRI and d) NTI with the effect of climate variables taken into account using multiple regression models that included all climate variables. Purple shading represents higher PCS values (PCS > 0) and, hence, clustered community structure, whereas red shading represents lower PCS (PCS < 0) values and, hence, overdispersed community structure.

## PCS and spatiotemporal patterns

Previous studies examining spatial patterns of PCS for Angiosperms in the Northern Hemi-sphere reported results consistent with our findings, with more clustering in the north and

overdispersion towards the south [53, 54]. Our results for Angiosperms contradict previous descriptions of PCS patterns for Gymnosperms [25, 53] that have shown clustering in the southern and overdispersion in the northern regions of eastern North America. However, these findings support the contention that PCS varies with the clade under consideration.

The prevalence of clustered communities in certain regions of our study area could indicate a role for environmental filtering, such that the relatively harsher (e.g., cold, dry) environments in these locations tend to support closely related species assuming a strong match between traits that promote higher survival in harsh environments and phylogenetic relatedness. For instance, increased clustering in northeastern North America and the Great Plains could be the result of a north-south temperature filter and an east-west moisture filter, respectively. Similarly, shorter growing seasons might contribute to more phylogenetic clustering in the Appalachian Mountains. On the other hand, if competition is the primary driver creating these PCS patterns, we would expect closely related species to be competitively superior in harsh environments assuming a strong match between phylogenies and competitive ability. Testing which of these processes (environmental filtering or competition) is more important is difficult given that the relative importance of climate, biotic interactions, and other factors such as the dynamics of succession in determining these patterns remains unclear. Nevertheless, our observation of the same geographic patterns repeated across time could hint at common processes driving community assembly.

### Relationship between PCS and climate through time

Our results generally agree with previous studies in the effect of the different climate variables on PCS in that we found similar relationships, but with a much lower variance explained. For instance, a large amount of variation in PCS can usually be explained by temperature and precipitation variables [25, 26, 53, 55]. Although climate is known to have a central role in structuring communities when measured on the basis of species richness [10, 56] or beta diversity [57], our results suggest that it has a more limited role in explaining PCS (Table 1), at least at the coarse spatial and taxonomic scales of our study. While our findings regarding relationships between annual temperature and NRI differ from those reported for pine and podocarp clades based on OLS analysis [25], we intentionally removed gymnosperms from our analysis. Hence differences in the relationships between climate and NRI might be attributable to differential adaptations of angiosperms and gymnosperms to climate [58].

We found strong spatial autocorrelation in PCS and that accounting for this spatial structure increased the explanatory power of some climate variables, while also revealing non-significant relationships with other variables (i.e., mainly ETR and Pmax). Incorporating spatial effects also inverted the direction of the relationship for several variables (Tmax for NRI or Pmin for NTI; Table 1). The presence of strong spatial autocorrelation at local scales ($< 500m$) has been used to indicate that dispersal plays an important role in structuring these communities in the past. Since traits associated with dispersal and colonization are correlated with phylogeny, we would expect to see spatial autocorrelation in PCS at local scales. However, we are using pollen data to represent the composition of plant assemblages, which is more indicative of a regional signature and unlikely to reflect dispersal patterns at local scales.

For climate variables with a significant relationship with PCS, we found that climate-PCS relationships vary through time, especially when spatial autocorrelation is considered. Independently of spatial autocorrelation, PCS-climate relationships varied through time for all the environmental variables tested (Table 2). Changes in both intercepts and slopes through time suggest a signal of other factors, notably biotic interactions or dispersal lags, on PCS patterns. In terms of dispersal lags, previous studies have shown that past climate conditions and

associated dispersal limitations can influence present-day community composition and PCS. For instance, analyses of PCS of global conifer, angiosperm, and palm assemblages revealed that both past and current climate are important predictors for NRI [25, 55, 59]. Our study, however, highlights the changing importance of climate on PCS and calls for caution when interpreting the results from studies based on single-time periods. Taken together, our results suggest that the signal of climate on PCS of fossil pollen assemblages of Angiosperms in eastern North America is either (1) weak and inconsistent, and/or (2) obscured by the coarse spatial, temporal, and taxonomic resolution of the records themselves.

### Time-since-deglaciation and PCS

Our understanding of how disturbance influences PCS is mainly limited to studies analyzing relatively short time series [30–33, 35]. These studies suggest that communities that assemble post-disturbance initially tend to be phylogenetically clustered, with succession processes tending to increase overdispersion through time. Our results for NRI are consistent with this pattern for the most recent time periods (from ~9 ka BP to present) but deviate from it for more distant time periods (Fig 5), suggesting that the relationship between PCS and disturbance (here time-since deglaciation or DEGLAC) varies across time.

   Our model selection procedure showed the least support for a stable relationship between PCS and time-since-deglaciation. Instead, models that included temporal variation in both intercepts and slope were both preferred and had the most explanatory power. Temporal variation in model intercepts, but constant slopes would be consistent with temporal lags between deglaciation and changes in PCS over time as well as migration lags. Temporal variation in slopes in addition to intercepts indicates changes in the nature of the response of PCS to deglaciation. Relative to other regions (e.g., Europe), we might expect postglacial migration to be less constrained by dispersal barriers in North America where mountain ranges are aligned mainly north-south (notwithstanding the role of the Laurentian Great Lakes) leading to less consistent lags in response to deglaciation. In addition, recolonization of deglaciated regions by some species could have been driven by isolated patches of vegetation near the ice boundary rather than from distant locations [60]. This combined with the spatial autocorrelation we found in our data suggests that dispersal in response to deglaciation potentially influenced community composition and therefore PCS.

   This is also reflected in the patterns of PCS (particularly NRI) plotted over time and time-since-deglaciation. NRI displayed the expected pattern (clustered in sites that were relatively recently deglaciated) over the last 8000 to 9000 years where a majority of the glacial ice had already retreated. Deeper in time, we found deviations from this pattern where most sites were either phylogenetically overdispersed or had NRI close to 0. These averages are likely driven by sites that were not glaciated during the LGM thus giving them more time since a large disturbance event to become phylogenetically overdispersed.

### Caveats

It is important to consider several caveats when interpreting our results, the first of which relates to the coarse taxonomic, spatial, and temporal resolution of fossil pollen data. Our use of fossil pollen records typically are discernible only to the genus-level and represent a regional signal of vegetation. Although fossil pollen data are biased towards certain plant clades (i.e., wind pollinated tree taxa), they include genera that represent some of the most widespread and abundant plant taxa within the study region and therefore should characterize general vegetation patterns. However, the coarse taxonomic resolution of the fossil pollen data used in our analyses may impact PCS metrics due to a lack of terminal branches in the phylogenetic

tree. This may explain the differences in our PCS values when compared with species-level studies [26, 53] that performed analyses at a similar spatial scale as in this study. Genus-level data would likely tend to influence NTI more than NRI, given that NTI assigns greater emphasis to recent splits in the phylogeny than does NRI [17, 18]. Our use of genus-level data also may explain differences in the range of NRI and NTI values. The selected pool of taxa and the clade under consideration also could influence PCS patterns [26, 55]. Furthermore, most previous studies analyzed plot-based data at the species level, whereas we are analyzing records of plant taxa preserved in lake sediments, which may not represent local communities. For example, in a comparison of similarities between community patterns measured using plots versus pollen cores, [61] found that pollen samples best represented plant assemblages at spatial extents of 10 arcminutes and above. Our analyses also rely on paleoclimatic simulations from Earth System Models (ESMs), rather than observations of paleoclimate. ESMs possess known uncertainties, inaccuracies, and biases, and if the paleoclimate simulations exhibit systematic errors across space or during specific times, this could influence PCS-climate relationships. Lastly, a suite of other factors can influence plant-climate relationships and therefore PCS-climate-relationships, most notably atmospheric $CO_2$ concentration, which was much lower at the LGM, and megaherbivores, which were more abundant at the LGM.

## Conclusions

Our study reveals several insights into the spatiotemporal patterns of PCS and the relationship between PCS and environmental conditions. We found a consistent geographic pattern of PCS, but the ecological processes underlying these patterns cannot be fully discerned in this study. In addition, not only did we find spatial autocorrelation in PCS, but also that proper accounting of spatial autocorrelation altered the stability of PCS-climate relationships through time, suggesting that static PCS-climate relationships may have limited utility for making predictions under changing environments. This remains true for the relationship between PCS and time-since-deglaciation. We also observed that in the recent past, communities that initially established post-disturbance tended to be more clustered than those that succeeded them. Although relationships between PCS and environment have the potential to inform predictions of community phylogenetic characteristics as a response to climate, our study suggests that such predictions should be approached with caution.

## Supporting information

**S1 Fig. Phylogenetic tree with angiosperm (blue branches) and Gymnosperm taxa (green branches) from the Open Tree of Life for spermatophytes.** The tree is pruned to the taxa in the fossil pollen dataset. Node labels indicate estimated age of the node as in the 'datelife' database. Ages for the rest of the nodes were estimated using the bladj algorithm as implemented in the 'picante' package in R.
(TIFF)

**S2 Fig. Scatterplots for NRI against all climatic variables and time-since-deglaciation and regression lines for three OLS models.** Stable-Relationship (orange line), Stable-Slope (green line) and Changed-Relationship (blue line). Tmin = minimum temperature of the coldest month; Tmax = maximum temperature of the warmest month; Pmin = minimum precipitation of the driest month; Pmax = maximum precipitation of the wettest month; AET = mean yearly actual evapotranspiration; ETR = mean yearly ratio of actual and potential evapotranspiration; WDI = mean yearly water deficit index; DEGLAC = time-since-deglaciation.
(ZIP)

**S3 Fig. Scatterplots for NTI against all climatic variables and time-since-deglaciation and regression lines for three OLS models.** Stable-Relationship (orange line), Stable-Slope (green line), and Changed-Relationship (blue line). Note that each model is fitted to data for all time periods and hence a single adjusted $R^2$ and p-value for each model is presented in the same colors as the lines. Tmin = minimum temperature of the coldest month; Tmax = maximum temperature of the warmest month; Pmin = minimum precipitation of the driest month; Pmax = maximum precipitation of the wettest month; AET = mean yearly actual evapotranspiration; ETR = mean yearly ratio of actual and potential evapotranspiration; WDI = mean yearly water deficit index; DEGLAC = time-since-deglaciation.
(ZIP)

**S4 Fig. Relationship between net relatedness index (NRI) and three climate variables and their change through time according to the three fitted models and equal sample size through time (n = 12 in each time period).** Gray shading in the scatter plots represents the count of points falling in each bin (hexagons). The panels on the left (a, c, e) represent the overall relationship according to ordinary least square regression when pooling all time periods (Stable-Relationship; orange lines). The panels on the right (b, d, f) show the relationship between NRI and climate variables as estimated with the data for different time periods, with green lines representing Stable-Slope Model, blue lines representing Changed-Relationship Model, and the orange line showing the overall relationship from panels on the left for comparison (Stable-Relationship). Shaded areas represent the confidence intervals at 95% for the regression lines. Note that each model is fitted to data for all time periods and hence a single adjusted $R^2$ and p-value for each model is presented in the same colors as the lines.
(TIFF)

**S5 Fig. Model intercepts through time with equal sample size through time (n = 12 in each time period).** Intercepts of OLS and SAR models for each variable plotted across time for a) Stable-Slope and b) Changed-Relationship models. Tmin = minimum temperature of the coldest month; Tmax = maximum temperature of the warmest month; Pmin = minimum precipitation of the driest month; Pmax = maximum precipitation of the wettest month; AET = mean yearly actual evapotranspiration; ETR = mean yearly ratio of actual and potential evapotranspiration; WDI = mean yearly water deficit index; DEGLAC = time-since-deglaciation.
(TIFF)

**S6 Fig. Model slopes through time with equal sample size through time (n = 12 in each time period).** Slopes of OLS and SAR models for each variable plotted across time for the Changed-Relationship models. Note that a) contains slopes for ETR while b) shows the same for all other variables; this was separated for readability as the slope values for ETR varied at a bigger scale than that of other variables. Tmin = minimum temperature of the coldest month; Tmax = maximum temperature of the warmest month; Pmin = minimum precipitation of the driest month; Pmax = maximum precipitation of the wettest month; AET = mean yearly actual evapotranspiration; ETR = mean yearly ratio of actual and potential evapotranspiration; WDI = mean yearly water deficit index; DEGLAC = time-since-deglaciation.
(TIFF)

**S7 Fig. Pattern of time-since-deglaciation on net relatedness index (NRI) and nearest taxon index (NTI) with equal sample size through time.** Each cell in the heatmap represents the average a) NRI or b) NTI value for cells of a particular time-since-deglaciation at a particular time period. Black squares indicate absence of the particular time-since-deglaciation class for that time period. The second row of heatmaps represent the average of residuals for c) NRI and d) NTI with the effect of climate variables taken into account using multiple regression

models that included all climate variables. Blue colors represent higher PCS values (PCS > 0) and, hence, clustered community structure, whereas red colors represent lower PCS (PCS < 0) values and, hence, overdispersed community structure.
(TIFF)

**S1 Table. Spatial autocorrelation of raw NRI and NTI values.**
(DOCX)

**S2 Table. Moran's I values for residuals of OLS models relating NRI and NTI with all variables (geographic, climatic, and time-since-deglaciation).** Tmin = minimum temperature of the coldest month; Tmax = maximum temperature of the warmest month; Pmin = minimum precipitation of the driest month; Pmax = maximum precipitation of the wettest month; AET = mean yearly actual evapotranspiration; ETR = mean yearly ratio of actual and potential evapotranspiration; WDI = mean yearly water deficit index; DEGLAC = time-since-deglaciation.
(DOCX)

**S3 Table. Moran's I values for residuals of SAR models relating NRI and NTI with seven climate variables.** Moran's I values are based on models fitted with error-SAR at the specified distances and are all non-significant (ns). Tmin = minimum temperature of the coldest month; Tmax = maximum temperature of the warmest month; Pmin = minimum precipitation of the driest month; Pmax = maximum precipitation of the wettest month; AET = mean yearly actual evapotranspiration; ETR = mean yearly ratio of actual and potential evapotranspiration; WDI = mean yearly water deficit index; DEGLAC = time-since-deglaciation.
(DOCX)

**S4 Table. AIC values for OLS and SAR models relating NRI and NTI with seven climate variables.** AIC values of SAR are based on models fitted with error-SAR at the specified distances. The lowest AIC values for each climatic variable and PCS metric (i.e., NRI and NTI) is highlighted in bold. Tmin = minimum temperature of the coldest month; Tmax = maximum temperature of the warmest month; Pmin = minimum precipitation of the driest month; Pmax = maximum precipitation of the wettest month; AET = mean yearly actual evapotranspiration; ETR = mean yearly ratio of actual and potential evapotranspiration; WDI = mean yearly water deficit index; DEGLAC = time-since-deglaciation.
(DOCX)

**S5 Table. Parameters of ordinary least square regression (OLS) and spatial autoregressive (SAR) models relating two metrics of phylogenetic community structure (PCS; net relatedness index—NRI—And nearest taxon index—NTI) with seven climate variables and equal sample size through time (n = 12 in each time period).** Each combination of PCS metric and climate variable was modeled three times, allowing distinct levels of variation to study the evolution of the parameters through time: stable-relationship, stable-slope, and changed relationship. SAR models reported here were fit selecting neighbors at distances of 480 km for NRI and NTI. Tmin = minimum temperature of the coldest month; Tmax = maximum temperature of the warmest month; Pmin = minimum precipitation of the driest month; Pmax = maximum precipitation of the wettest month; AET = mean yearly actual evapotranspiration; ETR = mean yearly ratio of actual and potential evapotranspiration; WDI = mean yearly water deficit index; DEGLAC = time-since-deglaciation.
(DOCX)

**S6 Table. ANOVA-based model selection of ordinary least square regression (OLS) and spatial autoregressive (SAR) models relating net relatedness index (NRI) and nearest**

**taxon index (NTI) with seven climate variables and equal sample size through time (n = 12 in each time period).** Each ANOVA was run for each combination of PCS metric and climate variable comparing three different models that allow distinct levels of variation to study the evolution of the regression parameters through time: stable-relationship, stable-slope, and changed relationship. SAR models reported here were fit selecting neighbors at distances of 360 km. Tmin = minimum temperature of the coldest month; Tmax = maximum temperature of the warmest month; Pmin = minimum precipitation of the driest month; Pmax = maximum precipitation of the wettest month; AET = mean yearly actual evapotranspiration; ETR = mean yearly ratio of actual and potential evapotranspiration; WDI = mean yearly water deficit index; DEGLAC = time-since-deglaciation.
(DOCX)

## Acknowledgments

We thank the contributions of several anonymous reviewers whose comments greatly improved this manuscript.

## Author Contributions

**Conceptualization:** Kavya Pradhan, Diego Nieto-Lugilde, Matthew C. Fitzpatrick.

**Data curation:** Matthew C. Fitzpatrick.

**Formal analysis:** Kavya Pradhan, Diego Nieto-Lugilde.

**Funding acquisition:** Matthew C. Fitzpatrick.

**Methodology:** Diego Nieto-Lugilde, Matthew C. Fitzpatrick.

**Project administration:** Matthew C. Fitzpatrick.

**Resources:** Matthew C. Fitzpatrick.

**Supervision:** Diego Nieto-Lugilde, Matthew C. Fitzpatrick.

**Visualization:** Diego Nieto-Lugilde.

**Writing – original draft:** Kavya Pradhan, Diego Nieto-Lugilde.

**Writing – review & editing:** Kavya Pradhan, Diego Nieto-Lugilde, Matthew C. Fitzpatrick.

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
