## [Decision Letter · Decision Letter 0]

5 Jan 2021

PONE-D-20-31298

Relationships between climate and phylogenetic community structure of fossil pollen assemblages are not constant during the last deglaciation

PLOS ONE

Dear Dr. Fitzpatrick,

Thank you for submitting your manuscript to PLOS ONE. After careful consideration, we feel that it has merit but does not fully meet PLOS ONE’s publication criteria as it currently stands. Therefore, we invite you to submit a revised version of the manuscript that addresses the points raised during the review process.

We look forward to receiving your revised manuscript.

Kind regards,

Ji-Zhong Wan

Academic Editor

PLOS ONE

Journal Requirements:

2) We note that you have included the phrase “data not shown” in your manuscript. Unfortunately, this does not meet our data sharing requirements. PLOS does not permit references to inaccessible data. We require that authors provide all relevant data within the paper, Supporting Information files, or in an acceptable, public repository. Please add a citation to support this phrase or upload the data that corresponds with these findings to a stable repository (such as Figshare or Dryad) and provide and URLs, DOIs, or accession numbers that may be used to access these data. Or, if the data are not a core part of the research being presented in your study, we ask that you remove the phrase that refers to these data.

Reviewers' comments:

Reviewer's Responses to Questions

**Comments to the Author**

1. Is the manuscript technically sound, and do the data support the conclusions?

Reviewer #1: Yes

Reviewer #2: Yes

2. Has the statistical analysis been performed appropriately and rigorously? 

Reviewer #1: Yes

Reviewer #2: Yes

3. Have the authors made all data underlying the findings in their manuscript fully available?

Reviewer #1: Yes

Reviewer #2: Yes

4. Is the manuscript presented in an intelligible fashion and written in standard English?

Reviewer #1: Yes

Reviewer #2: Yes

5. Review Comments to the Author

Reviewer #1: This is an interesting and well written manuscript, using a Phylogenetic Community Structure approach to understand palaeoecological changes in North America. The authors find a spatial trend of phylogenetic clustering in the northeast and "overdispersion" in the southwestern study region. This papers offers new views for the palaeoecological community, which has not used phylogeny for environmental reconstruction. I have few specific comments:

Page 5 the authors state that: Lastly, most previous studies analyzed plot-based data at the species level,

whereas we are analyzing records of taxa preserved in lake sediments, which may not represent local communities.

Well, first the authors mentioned that in this study only angiopserm trees were used. And what do the authors understand by local communities? shore lake communities? For example, boreal forest communities at many lakes could be considered local, however, they could generate a regional pollen signal.

In a more general sense, the authors should be more specific how their approach could improve present paleoecological reconstructions like MAT (modern analogue techniques) or other available techniques.

Reviewer #2: There are no line numbers, and the page numbers re-start halfway through, which makes it difficult to provide comments.

This is a generally well-written and well-executed manuscript. However, I think the main conclusions and results change within the results and conclusion, and the main message is not consistent, as if different parts of the manuscript has been altered at different times. See my last specific comment, for p 8. Also the figures and legends are different in some cases. The take-home message would strengthen with a final revision.

The introduction discussed post-glacial migration/dispersal, and how it could affect species distributions and thus PCS. However, in the rest of the ms this is not much discussed. How could one account for dispersal? It seems that also spatial autocorrelation could be related to dispersal, when climate has been taken into account.

Also, the introduction talks about how environmental filtering and competition could affect PCS, but this is not mentioned when discussing the results from deglaciation (fig 5).

One of the main conclusions is that PCS varies in time. However, I think that PCS values in most cases are quite constant in time (fig 2, most relationships are flat except when they have large confidence intervals). In fig 3 and 4, most of the intercepts look pretty constant, with the exception of temperatures in model 3. The intercepts seems to vary together over all climate variables, which maybe suggests that there is something beyond climate that is having a constant effect on PCS in each time period. Maybe time since deglaciation is having this effect, which is shown in fig 5 to influence PCS.

Explain the three main statistical models a bit more in detail, and how to interpret them biologically. Why were these three models chosen? Why not build one model with all climate variable? What does the intercept mean biologically?(Isn’t analysing the intercept without the slope strange?, as in fig 3) Improving this explanation would help the reader to interpret the results and discussion.

Were the statistical models carried out for each separate time period? Otherwise, how can the intercept change in time? I must be misunderstanding something.

To be able to visualise changes in PCS, the maps in fig 1 could instead be spatially interpolated (kriging). This would remove some of the variability and allow readers to see the main trends.

Specific comments

p4 ‘(co-occurring species’ - no closing parentheses, twice.

p10 Community data matrix, missing preposition?

p12 first paragraph. What statistical analysis is this? Is it a regression against the latitude and longitude?

Fig 1. Maybe the black border around each dot could be removed, to show the color value better.

Fig 2 The right-hand panels are small and difficult to distinguish. Maybe the panel sub-titles (BP) could be removed, stating that the layout is the same as in fig 1 (although it’s not, as this contains data from 0 BP). The data points are difficult to see, and are covered by the confidence interval (which could be transparent). I cannot see the red line.

Fig 3,4 What do the horizontal dashed lines mean? Why are only some variables plotted in fig 4?

p16. ‘Similar to OLS models’ - has this result been presented?

p2 ‘both PCS metrics showed a positive relationship with DEGLAC (Table 1).’ The values for slopes for DEGLAC and the first two models are zero in table 1? Slope values are not presented for the third model.

Fig 5 legend should include both NTI and NRI. Grey squares = black squares. Explain the residuals (lower row).

p4 ‘Finally, we also found that the relationship between time-since-deglaciation and PCS also has not been constant through time.’ I thought that fig 5 shows that the relationship deglac and PCS has followed a general trend, so this is a little unclear. As stated on p 3: ‘we found phylogenetically clustered communities (positive PCS metrics) in sites that had deglaciated more recently.’ See next comment.

p4 ‘Angiosperm assemblages tended to become more phylogenetically clustered in the northeastern parts of the study area and more overdispersed in the southwestern regions.’ This is probably related to fig 5, as recently deglaciated areas have positive PCS. See previous comment.

p7 ‘climate on PCS either is (1) weak and inconsistent in fossil pollen assemblages of Angiosperms in eastern North America since the LGM, and/or’ could be ‘climate on PCS in fossil pollen assemblages of Angiosperms in eastern North America since the LGM is either (1) weak and inconsistent, and/or’

p7 ‘We show that the relationship between PCS and time-since-deglaciation (DEGLAC) is also unstable across time.’ I think that ‘unstable’ might not be the right word? Unstable implies that it is random, whereas e.g. ’varying’ implies that it is caused by something. Even though the results can’t prove any causes, they could use the word varying instead. Unless the low statistical power (r2, table 1) implies that the results are random.

p8 ‘Although relationships between PCS and environment have the potential to inform predictions, our study suggests that such predictions should be approached with caution.’ Vague statement. What kind of predictions? I think a more interesting conclusion is how climate, deglaciation and migration lags affect PCS.

p8 ‘cannot be discerned.’ add ‘in this analysis/study’. Also later in the conclusion it is stated ‘communities that initially establish post-disturbance tend to be more dispersed than those that proceed them, but this pattern was not consistent’ which suggests that the ecological processes underlying these patterns could be discerned.

6. PLOS authors have the option to publish the peer review history of their article (what does this mean?). If published, this will include your full peer review and any attached files.

Reviewer #1: No

Reviewer #2: No

---

## [Author Response · Author response to Decision Letter 0]

19 Feb 2021

COMMENTS FROM EDITOR

We note that you have included the phrase “data not shown” in your manuscript. Unfortunately, this does not meet our data sharing requirements. PLOS does not permit references to inaccessible data. We require that authors provide all relevant data within the paper, Supporting Information files, or in an acceptable, public repository. Please add a citation to support this phrase or upload the data that corresponds with these findings to a stable repository (such as Figshare or Dryad) and provide and URLs, DOIs, or accession numbers that may be used to access these data. Or, if the data are not a core part of the research being presented in your study, we ask that you remove the phrase that refers to these data.

ANSWER: This was a typo and should have read “results not shown”. Nonetheless, all results and data presented in the revised manuscript are now provided, either in the supplement or via web links.

COMMENTS FROM REVIEWER #1

This is an interesting and well written manuscript, using a Phylogenetic Community Structure approach to understand palaeoecological changes in North America. The authors find a spatial trend of phylogenetic clustering in the northeast and "overdispersion" in the southwestern study region. This papers offers new views for the palaeoecological community, which has not used phylogeny for environmental reconstruction. I have few specific comments:

ANSWER: We thank the reviewer for their detailed review and useful comments that greatly improved the manuscript. We hope that we have fully addressed all concerns. We also note that it was not our intention to inform environmental reconstruction, but rather to examine relationships between PCS and climate through time and across space.

Page 5 the authors state that: Lastly, most previous studies analyzed plot-based data at the species level, whereas we are analyzing records of taxa preserved in lake sediments, which may not represent local communities.

Well, first the authors mentioned that in this study only angiopserm trees were used. And what do the authors understand by local communities? shore lake communities? For example, boreal forest communities at many lakes could be considered local, however, they could generate a regional pollen signal.

ANSWER: We have edited the Discussion to address this point. In particular, we now have a section labeled “Caveats” that discusses possible challenges with using fossil pollen records to represent vegetation assemblages. We have also edited the text in question to read: “Lastly, most previous studies analyzed plot-based data at the species level, whereas we are analyzing records of plant taxa preserved in lake sediments, which may reflect a regional signal of vegetation composition.”

In a more general sense, the authors should be more specific how their approach could improve present paleoecological reconstructions like MAT (modern analogue techniques) or other available techniques.

ANSWER: We appreciate this comment from the reviewer, but feel that discussions of paleoecological reconstructions are beyond the scope of the current study.

COMMENTS FROM REVIEWER #2

There are no line numbers, and the page numbers re-start halfway through, which makes it difficult to provide comments.

ANSWER: Sorry for any confusion, this was an issue with google docs. We have fixed the page numbering issue and added line numbers. 

This is a generally well-written and well-executed manuscript. However, I think the main conclusions and results change within the results and conclusion, and the main message is not consistent, as if different parts of the manuscript has been altered at different times. See my last specific comment, for p 8. Also the figures and legends are different in some cases. The take-home message would strengthen with a final revision.

ANSWER: Thank you for this insightful comment. We have thoroughly edited the text, figures, and figure captions for consistency. 

The introduction discussed post-glacial migration/dispersal, and how it could affect species distributions and thus PCS. However, in the rest of the ms this is not much discussed. How could one account for dispersal? It seems that also spatial autocorrelation could be related to dispersal, when climate has been taken into account.

ANSWER: We have edited the Discussion to make explicit the link between time-since-deglaciation (DEGLAC) and spatial autocorrelation with dispersion.

Also, the introduction talks about how environmental filtering and competition could affect PCS, but this is not mentioned when discussing the results from deglaciation (fig 5).

ANSWER: Similarly, we edited the Discussion to connect environmental filtering and competition to PCS in the section dedicated to PCS-DEGLAC.

One of the main conclusions is that PCS varies in time. However, I think that PCS values in most cases are quite constant in time (fig 2, most relationships are flat except when they have large confidence intervals). In fig 3 and 4, most of the intercepts look pretty constant, with the exception of temperatures in model 3. The intercepts seems to vary together over all climate variables, which maybe suggests that there is something beyond climate that is having a constant effect on PCS in each time period. Maybe time since deglaciation is having this effect, which is shown in fig 5 to influence PCS.

ANSWER: Thank you for the comment! We agree that there were some inconsistencies in how we wrote about our results. Our main conclusion is that PCS-climate relationships vary over time while geographic patterns of PCS are mostly stable (as seen in FIg 1). We edited our results and discussion to clarify. As for the second part of your comment regarding the intercepts and how they vary together, our thoughts were along the same lines, i.e, something other than climate (we also thought about deglaciation) may be causing this pattern. However, Fig 3 and 4 also include DEGLAC and show that it changes in the same pattern which did not support our hypothesis that DEGLAC was having this effect. Without additional study, we are hesitant to speculate further. 

Explain the three main statistical models a bit more in detail, and how to interpret them biologically. Why were these three models chosen? Why not build one model with all climate variable? What does the intercept mean biologically?(Isn’t analysing the intercept without the slope strange?, as in fig 3) Improving this explanation would help the reader to interpret the results and discussion. 

ANSWER: We edited our methods section (3rd paragraph under Analysis) to address concerns regarding interpretation of the three models, our reasoning behind selecting them, and modeling individual variables rather than creating a multiple regression. 

Were the statistical models carried out for each separate time period? Otherwise, how can the intercept change in time? I must be misunderstanding something.

ANSWER: Sorry if this wasn’t clear. In models 2 and 3, time is included as an additional variable (as a fixed effect variable in model 2 and as a random effect variable in model 3). We have clarified these points in the manuscript: “In the Stable-Relationship models, time is neglected, whereas in the Stable-Slope models, time is included as an additional variable. In the Changed-Relationship models, time is included as a variable interacting with climate.”.

To be able to visualise changes in PCS, the maps in fig 1 could instead be spatially interpolated (kriging). This would remove some of the variability and allow readers to see the main trends.

ANSWER: Thank you for this great suggestion. We have used Inverse Distance Weighted interpolation instead of kriging. The reason is that Kriging was more computationally challenging and IDW provided an suitable alternative to improve data visualization.

Specific comments

p4 ‘(co-occurring species’ - no closing parentheses, twice.

ANSWER: Sorry for any confusion. We have added the missing closing parentheses. 

p10 Community data matrix, missing preposition?

ANSWER: Good catch. Now reads “the community data matrix”.

p12 first paragraph. What statistical analysis is this? Is it a regression against the latitude and longitude?

ANSWER: We added text to clarify that this is ordinary least squares regressions of NRI and NTI against latitude and longitude.

Fig 1. Maybe the black border around each dot could be removed, to show the color value better.

ANSWER: Thank you for this suggestion. We tried recreating the figures without the black border around each point & found that the pattern is easier to see with the black border, which adds contrast. In any case, this figure has changed with IDW interpolation to improve data visualization, as suggested above.

Fig 2 The right-hand panels are small and difficult to distinguish. Maybe the panel sub-titles (BP) could be removed, stating that the layout is the same as in fig 1 (although it’s not, as this contains data from 0 BP). The data points are difficult to see, and are covered by the confidence interval (which could be transparent). I cannot see the red line.

ANSWER: We edited Fig 2 to improve visibility of the right-hand panels. We reduced the number of time-slices that we display in this figure and opted to retain the confidence intervals as now the dots are more visible than in the previous version of this figure. 

Apologies for the confusion about the “red line”. We edited the caption as this was an error and we meant “orange line” instead, which is displayed in the left-hand panels.

Fig 3,4 What do the horizontal dashed lines mean? 

ANSWER: We removed the lines since they were not relevant to the interpretation of our results

Why are only some variables plotted in fig 4?

ANSWER: All variables are plotted in fig4. However, the reason was not clear in the figure caption. We have clarified this by adding the following text: Note that a) contains slopes for ETR while b) shows the same for all other variables; this was separated for readability as the magnitude of slope values for ETR was much greater than that of other variables.”

p16. ‘Similar to OLS models’ - has this result been presented?

ANSWER: Yes, the results for OLS models are presented at the beginning of this section, which states: “Most OLS models relating NRI or NTI with climate variables were significant (Table 1).” Sorry if this was not clear.

p2 ‘both PCS metrics showed a positive relationship with DEGLAC (Table 1).’ The values for slopes for DEGLAC and the first two models are zero in table 1? Slope values are not presented for the third model.

ANSWER: Sorry for any confusion. This text was from a previous version of the manuscript and has been deleted.

Fig 5 legend should include both NTI and NRI. Grey squares = black squares. Explain the residuals (lower row).

ANSWER: Thank you for catching that! We have edited the legend to include both NRI and NTI, changed “grey squares” to “black squares” and added a section explaining the lower row

p4 ‘Finally, we also found that the relationship between time-since-deglaciation and PCS also has not been constant through time.’ I thought that fig 5 shows that the relationship deglac and PCS has followed a general trend, so this is a little unclear. 

ANSWER: Thank you for pointing this out as it highlighted lack of clarity in our writing. We found that time-since-deglaciation and PCS (in particular NRI) follow a general trend (more recently deglaciated sites have clustered communities), but only between 0BP and around 9000BP. Beyond that, we cannot extend this generalization (Fig 5). This agrees with the results from NRI-DEGLAC models in that the relationship between NRI and DEGLAC changes through time. Patterns in Fig 5 for NTI are much more variable and cannot be generalized. We have added text in both the results and discussion to clarify.

As stated on p 3: ‘we found phylogenetically clustered communities (positive PCS metrics) in sites that had deglaciated more recently.’ See next comment.

ANSWER: Figure 5 shows that the patterns vary across time and supports the results from PCS-DEGLAC model selection of the Changed-Relationship model. In figure 5, the clustered communities are in sites that deglaciated more recently, but this is only the case for the recent past. When we look further in time, the pattern is not as clear (we instead see more sites that are either overdispersed or around 0). Our original sections on PCS-DEGLAC for both the results and the discussion were not clearly written and did not contain enough detail to explain this. Thank you for pointing out this vagueness! We have rewritten both sections to clarify these results based on this comment.

p4 ‘Angiosperm assemblages tended to become more phylogenetically clustered in the northeastern parts of the study area and more overdispersed in the southwestern regions.’ This is probably related to fig 5, as recently deglaciated areas have positive PCS. See previous comment.

ANSWER: We agree that there is a connection between deglaciation and the PCS patterns such that sites in the south that were not glaciated are the ones that tend to be overdispersed. To connect the two more explicitly, we added text in our discussion (Time-since-deglaciation and PCS section) 

p7 ‘climate on PCS either is (1) weak and inconsistent in fossil pollen assemblages of Angiosperms in eastern North America since the LGM, and/or’ could be ‘climate on PCS in fossil pollen assemblages of Angiosperms in eastern North America since the LGM is either (1) weak and inconsistent, and/or’

ANSWER: We edited the text as suggested.

p7 ‘We show that the relationship between PCS and time-since-deglaciation (DEGLAC) is also unstable across time.’ I think that ‘unstable’ might not be the right word? Unstable implies that it is random, whereas e.g. ’varying’ implies that it is caused by something. Even though the results can’t prove any causes, they could use the word varying instead. Unless the low statistical power (r2, table 1) implies that the results are random.

ANSWER: We agree that “varying” is a better replacement for “unstable” and have made changes in the manuscript accordingly.

p8 ‘Although relationships between PCS and environment have the potential to inform predictions, our study suggests that such predictions should be approached with caution.’ Vague statement. What kind of predictions? I think a more interesting conclusion is how climate, deglaciation and migration lags affect PCS.

ANSWER: We have added “predictions of community phylogenetic characteristics as a response to climate” to clarify the kind of predictions we are referring to. 

p8 ‘cannot be discerned.’ add ‘in this analysis/study’. 

Answer: Fixed. We added the suggested text.

Also later in the conclusion it is stated ‘communities that initially establish post-disturbance tend to be more dispersed than those that proceed them, but this pattern was not consistent’ which suggests that the ecological processes underlying these patterns could be discerned.

ANSWER: We are not sure we follow the Reviewer's comment. It is true that in some cases we found communities that initially establish post-disturbance tend to be more clustered than those that proceed them, but we also state that this pattern was not consistent and therefore made it challenging to draw any firm conclusions regarding the ecological processes driving these patterns & we are hesitant to speculate otherwise. Note: We also changed the text from “dispersed” to “clustered” as that was an error on our part.

---

## [Decision Letter · Decision Letter 1]

18 May 2021

PONE-D-20-31298R1

Relationships between climate and phylogenetic community structure of fossil pollen assemblages are not constant during the last deglaciation

PLOS ONE

Dear Dr. Fitzpatrick,

Thank you for submitting your manuscript to PLOS ONE. After careful consideration, we feel that it has merit but does not fully meet PLOS ONE’s publication criteria as it currently stands. Therefore, we invite you to submit a revised version of the manuscript that addresses the points raised during the review process.

*I'm awfully sorry it took so long to get back to you. I was recently assigned as Editor for your manuscript at the R1 stage and seeked for another review (reviewer #4) prior to taking a decision.*

*As you will see, all reviewers were pretty positive, but reviewer #4 noted that the results are a bit difficult to interpret and shared some thoughts you may consider to improve the discussion and interpretation.*

We look forward to receiving your revised manuscript.

Kind regards,

Walter Finsinger, PhD

Academic Editor

PLOS ONE

Journal Requirements:

Reviewers' comments:

Reviewer's Responses to Questions

**Comments to the Author**

1. If the authors have adequately addressed your comments raised in a previous round of review and you feel that this manuscript is now acceptable for publication, you may indicate that here to bypass the “Comments to the Author” section, enter your conflict of interest statement in the “Confidential to Editor” section, and submit your "Accept" recommendation.

Reviewer #2: All comments have been addressed

Reviewer #3: All comments have been addressed

Reviewer #4: (No Response)

2. Is the manuscript technically sound, and do the data support the conclusions?

Reviewer #2: Yes

Reviewer #3: Yes

Reviewer #4: Yes

3. Has the statistical analysis been performed appropriately and rigorously? 

Reviewer #2: Yes

Reviewer #3: Yes

Reviewer #4: Yes

4. Have the authors made all data underlying the findings in their manuscript fully available?

Reviewer #2: Yes

Reviewer #3: Yes

Reviewer #4: Yes

5. Is the manuscript presented in an intelligible fashion and written in standard English?

Reviewer #2: Yes

Reviewer #3: Yes

Reviewer #4: Yes

6. Review Comments to the Author

Reviewer #2: Thank you for revising your manuscript, I think it has greatly improved (but there are still no line numbers).

Reviewer #3: This topic is interesting, the application prospect of this technique is also feasible, and the data in the manuscript is reliable

Reviewer #4: I see that the manuscript is at the R1 stage; I was not involved in the first round of reviews. The author responses to reviewers are thoughtful and the manuscript is written well. On a point raised by Reviewer1, I agree with the authors that discussing the implications of these analyses for pollen-based environmental interpretations is outside the scope of this analysis.

This paper carries value as one of the first mappings of phylogenetic tree data onto pollen data, which is non-trivial given the taxonomic ambiguities in pollen data. This paper makes the useful finding that PCS-climate relationships are not constant through time. All analytical and statistical methods seem basically sound to me; the pollen data handling relies on approaches developed in prior papers and the phylogenetic and spatial modeling analyses seem sound. The results are a bit difficult to interpret, but the paper does a good job of presenting conclusions that are relatively solid and noting caveats where appropriate.

A couple of thoughts that might be useful for the discussion and interpretation. First, most of the analyses and hypotheses are framed around assumptions that temperature is the primary environmental filter, and so assume that environmental filtering will be strongest in the north. However, in North America there are broadly two environmental gradients/filters: a north-south temperature filter and an east-west moisture filter. This second filter would help explain the pattern of phylogenetic clustering in the far western part of the study domain, where environmental filtering is strongest due to scarce water availability.

Second, it seems striking to me that many of the overdispersed loci are in the central US, roughly at

The continental ecotone between the Great Plains to the west and the mesic deciduous forests to the east. So, I wonder if some of the overdispersion comes at this physiognomic ecotone (grasses/herbs to the west, trees to the east) that I suspect also represents a phylogenetic ecotone. If so, perhaps one could invoke ecotones as a previously underrecognized reason for phylogenetic overdispersion.

For the Caveats section, add the point that these analyses rely upon paleoclimatic simulations from Earth System Models, which carry known uncertainties and inaccuracies. If the ESMs are systematically wrong in some places, this could lead to some of the apparent shifts in PCS-climate relationships. Could also note that CO2 was much lower at the LGM and more diverse suites of megaherbivores were present, both of which might have altered plant-climate (and PCS-climate) relationships.

As an aside, two issues in the manuscript created unnecessary work for this volunteer peer reviewer. First, no line numbers were available, which makes it hard to precisely pinpoint comments. Second, the figure legends were scattered throughout the main text, while the figures were all at the end. This made it quite hard to find the matching legends and figures in my PDF. So, in the future, please put legends and figures all in one place. (I personally prefer all at the end of the ms., but others prefer embedding figures and text in the ms.)

Other line-by-line comments:

For Figure 2, include in the panels a R2, p-value, or other measure of goodness of fit or significance.

Figure 4: what are the units for the slopes on the y-axis?

P6 ‘Extensive paleoecological time series’ – extensive in time and/or space? Clarify.

P6: Good statement of expectations at the bottom of P6. Note too that one might expect stronger environmental filtering in the semi-arid climates of the Great Plains region.

P7: This review of Ambrosia is a bit unclear and also omits Ambrosia’s resemblance to Iva.

P12: The models all seem fine but the review of models 2 & 3 both invoke lags as a reason for changes in either the intercept of slope. I strongly recommend removing any mention of lags at this point, because there are multiple reasons for why PCS-climate relationships might change over time.

P14 Fig 2 legend – use the ‘Stable-Relationship’ model terminology when referring to the leftmost plots, for consistency with Methods.

P24 Capitalize Northern Hemisphere

P27 times periods -> time periods

7. PLOS authors have the option to publish the peer review history of their article (what does this mean?). If published, this will include your full peer review and any attached files.

Reviewer #2: No

Reviewer #3: **Yes: **Dr. Dharmendra

Reviewer #4: No

---

## [Editor Report · Decision Letter 2]

16 Jun 2021

Relationships between climate and phylogenetic community structure of fossil pollen assemblages are not constant during the last deglaciation

PONE-D-20-31298R2

Dear Dr. Fitzpatrick,

We’re pleased to inform you that your manuscript has been judged scientifically suitable for publication and will be formally accepted for publication once it meets all outstanding technical requirements.

Kind regards,

Walter Finsinger, PhD

Academic Editor

PLOS ONE

Additional Editor Comments (optional):

None

Reviewers' comments:

None

---

## [Editor Report · Acceptance letter]

25 Jun 2021

PONE-D-20-31298R2 

Relationships between climate and phylogenetic community structure of fossil pollen assemblages are not constant during the last deglaciation 

Dear Dr. Fitzpatrick:

I'm pleased to inform you that your manuscript has been deemed suitable for publication in PLOS ONE. Congratulations! Your manuscript is now with our production department. 

Kind regards, 

on behalf of

Dr. Walter Finsinger 

Academic Editor

PLOS ONE